# Next generation biosecurity: Towards genome based identification to prevent spread of agronomic pests and pathogens using nanopore sequencing

**Jürg E. Frey**[1], **Beatrice Frey**[1], **Daniel Frei**[1], **Simon Blaser**[2,3], **Morgan Gueuning**[1¤], **Andreas Bühlmann**[4]*

1 Research Group Molecular Diagnostics Genomics and Bioinformatics, Department of Method Development and Analytics, Agroscope, Wädenswil, Switzerland, 2 Department of Plants and Plant Products, Agroscope Phytosanitary Service, Agroscope, Wädenswil, Switzerland, 3 Forest Health and Biotic Interactions, Federal Research Station for Forest, Snow and Landscape, Birmensdorf, Switzerland, 4 Strategic Research Division Food Microbial Systems, Research Group Product Quality and –Innovation, Agroscope, Wädenswil, Switzerland

¤ Current address: Department of Research and Development, Blood Transfusion Service Zurich, Swiss Red Cross, Schlieren, Switzerland
* Andreas.buehlmann@agroscope.admin.ch

**Data Availability Statement:** All sequence data generated are available on NCBI Sequence Read Archive (accession number PRJNA783774).

## Abstract

The unintentional movement of agronomic pests and pathogens is steadily increasing due to the intensification of global trade. Being able to identify accurately and rapidly early stages of an invasion is critical for developing successful eradication or management strategies. For most invasive organisms, molecular diagnostics is today the method of choice for species identification. However, the currently implemented tools are often developed for certain taxa and need to be adapted for new species, making them ill-suited to cope with the current constant increase in new invasive species. To alleviate this impediment, we developed a fast and accurate sequencing tool allowing to modularly obtain genetic information at different taxonomical levels. Using whole genome amplification (WGA) followed by Oxford nanopore MinION sequencing, our workflow does not require any a priori knowledge on the investigated species and its classification. While mainly focusing on harmful plant pathogenic insects, we also demonstrate the suitability of our workflow for the molecular identification of bacteria (*Erwinia amylovora* and *Escherichia coli*), fungi (*Cladosporium herbarum*, *Colletotrichum salicis*, *Neofabraea alba*) and nematodes (*Globodera rostochiensis*). On average, the pairwise identity between the generated consensus sequences and best GenBank BLAST matches was 99.6 ± 0.6%. Additionally, assessing the generated insect genomic dataset, the potential power of the workflow to detect pesticide resistance genes, as well as arthropod-infecting viruses and endosymbiotic bacteria is demonstrated.

## Introduction

Agronomic pests and pathogens cause enormous economic losses to global food production [1–3] and the rate of their unintentional transportation along global trading chains is steadily

**Funding:** The author(s) received no specific funding for this work.

**Competing interests:** The authors have declared that no competing interests exist.

increasing [4]. In order to prevent spread and to apply appropriate pest control strategies, accurate and early identification is key and represents therefore an important pillar for sustainable food production system [5, 6].

First described in 2003 [7], DNA barcoding has become a widely used standard diagnostic tool for molecular species identification [8]. The method relies on amplification and sequencing of a short DNA 'barcode' fragment, which is subsequently queried against a publicly available reference database such as 'Barcoding of Life Database' [9], EPPO-Q-bank (https://qbank. eppo.int/) [10] or NCBI GenBank [11]. The closest matching reference record is used for species identification, provided the barcode fragment is similar enough and that there is a gap between the match to conspecific specimen versus others [7, 12]. The first standardized barcode described was an approximately 658 base pairs long fragment from the 3' end of the mitochondrial cytochrome *c* oxidase I (COI) gene, which is used for the identification of specimens from the animal kingdom [7, 12]. The molecular identification of bacterial and fungal pathogens relies on other genetic markers including the ribosomal RNA 16s [12] or the internal transcribed spacer ITS [13], respectively. Due to its generic workflow, DNA barcoding is nowadays applied for the identification of genetically described plant pathogens and pests from a broad range of different taxa [8, 12].

In DNA barcoding, the diagnostic gene fragment is amplified by polymerase chain reaction (PCR) using a pair of primers matching regions flanking the targeted diagnostic 'barcode' fragment [7]. However, genetic variation between and within species may alter primer affinities and cause PCR failure if mismatches occur at primer binding sites [14]. To avoid allelic dropout it is therefore crucial to have a-priori information on potential intraspecific variation of the targeted species. Although essential, this information is too often lacking for diagnostic tasks where new, genetically undescribed pest and pathogen species or biotypes arrive on traded commodities (e. g. plant products) from many different geographical regions [8]. A second hurdle of DNA barcoding is that in some cases the short barcoding fragment does not harbor enough genetic variation to allow distinguishing closely related species. As an example, several harmful quarantine insects species, mainly those of relatively recent evolutionary origin such as Tephritidae, cannot be differentiated at the species level using COI as the only genetic marker [15]. These problems were recognized before, leading to the suggestion to expand beyond single marker diagnostics into genome based diagnostics [16].

To address the shortcomings of traditional DNA barcoding for diagnostic purposes described above, we designed a generic workflow based on whole genome amplification (WGA) [17] and Oxford Nanopore MinION sequencing [18]. The subsequent data analysis relies on the commercial software program Geneious Prime v2021.1.1 (https://www.geneious. com/; Biomatters, New Zealand), using a custom-built workflow to automatize the sequence analysis and identification steps with minimal user intervention. In brief, (i) the processed sequencing reads are mapped to a custom-made reference database, (ii) among the contigs with the highest percentage of reference sequence covered, the one with the highest number of reads is further processed and (iii) queried against a local copy of the NCBI GenBank [11]. Using this diagnostic workflow, (i) no a-priori genetic information about primer binding sites of the target organism is needed and (ii) genetic data covering the entire genome are collected, enabling species identification using single or a combination of multiple genetic markers for which reliable sequence information is available or can be established.

In the present study, the developed workflow was successfully evaluated for the identification of mainly plant pathogenic organisms focusing on insect pests. By this, 66 insects belonging to 29 species of 13 genera and 5 orders were analyzed. Using adjusted nucleic acid extraction methods, the same workflow was also shown to be suitable for the identification of

bacteria (*Erwinia amylovora* and *Escherichia coli*), fungi (*Cladosporium herbarum*, *Colletotrichum salicis*, *Neofabraea alba*) and nematodes (*Globodera rostochiensis*). Beyond its use for species identification, the generated insect genomic dataset was further exploited for its power to detect pesticide resistance genes as well as the presence of arthropod infecting viruses and endosymbiotic bacteria.

## Material and methods

This workflow may start with DNA or with RNA. Below, we report the process used for RNA and using a pipetting robot for several tasks. Detailed protocols starting with DNA extraction using commercial kits (S1 File) or using a custom proteinase K extraction, respectively, are described in the accompanying protocols.io protocol (dx.doi.org/10.17504/protocols.io. bx7nprme).

### Sample collection

Samples were obtained from various sources. The specimens of *Drosophila suzukii* (Diptera, Drosophilidae) originated from a laboratory population at Agroscope in Wädenswil; these samples were adult flies frozen at -20˚C. *Botanophila fugax*, *Delia florilega*, and *Delia radicum* (Diptera, Anthomyiidae) were collected in 2020 in the frame of the annual survey for vegetable pests using yellow pan traps positioned near Wädenswil, Switzerland. Most of the other samples were collected at Swiss airports on shipments of imported plant products by the Swiss Federal Plant Protection Service (SPPS; operated jointly by the Federal Office for Agriculture (FOAG) and the Federal Office for the Environment (FOEN)) between 2017 and 2020. *Liriomyza sativae* and *Liriomyza trifolii* (Diptera, Agromyzidae); *Bemisia tabaci* and *Trialeurodes vaporariorum* (Hemiptera, Aleyrodidae); *Aphis gossypii* (Hemiptera, Aphididae); *Leucinodes orbonalis* (Lepidoptera, Crambidae); *Helicoverpa armigera*, *Spodoptera exigua*, and *Spodoptera litura* (Lepidoptera, Noctuidae); *Hyalomorpha halys* (Hemiptera, Pentatomidae); *Pristiphora appendiculata* (Hymenoptera, Tenthredinidae); *Anastrepha obliqua*, *Bactrocera dorsalis*, *Bactrocera latifrons*, *Ceratitis anonae*, *Ceratitis capitata*, and *Zeugodacus cucurbitae* (Diptera, Tephritidae); *Thrips palmi* (Thysanoptera, Thripidae) were all stored in 70% EtOH either at room temperature or at 4˚C. The remaining specimens were collected by the Pest and Prevention Service of the city of Zurich. *Aedes japonicus* (Diptera, Culicidae), *Lasius neglegtus*, *Lasius niger*, *Plagiolepis pygmaea*, *Tapinoma magnum*, and *Tetramorium meridionale* (Hymenoptera, Formicidae) were also stored in 70% EtOH either at room temperature or at 4˚C. The *Erwinia amylovora* (Enterobacterales, Enterobacteriaceae) sample was isolated from a positively diagnosed Swiss apple orchard sample. DNA for *Escherichia coli* strain K-12 (Enterobacterales, Enterobacteriaceae) was purchased from Sigma (Sigma-Aldrich, St. Louis, MO, USA). The nematode *Globodera rostochiensis* specimens were reared on potato cv Desiree growing in soil: silver sand (1:2 v/v) mix in 1 L clay pots under greenhouse conditions (23/19 ˚C, 15/9 h day night cycle). Potato tubers were inoculated with 8'000 nematode eggs and after 10–12 weeks all potato shoots were cut and the clay pots were moved to a drying chamber (30˚C, 50% humidity). After 7 days, tubers were removed from the soil. Remaining soil and sand were kept drying for a total of 30 days. Dried soil was split into 250 mL samples and processed using an automated soil sample extractor (MEKU, Wennigsen, Germany). Symptomatic apple fruit of cultivar *Malus pumila* var. Golden Delicious grown in a research orchard in Wädenswil, Switzerland (47.220433, 8.666590) and stored in a research scale cold storage facility were collected and disease causing pathogen identified by lesion morphology and by DNA barcoding of the ITS2 region of the fungal ribosomal DNA using primers ITS3 and ITS4 [19].

## Sample preparation

For small insects (e.g., Thrips, Aphids, Drosophila) whole body were used. For larger specimens (e.g., larvae of Tephritids or Lepidoptera) max. 2 mg of tissue was used. Tissue disruption was done on a Retsch mixer mill MM200 (Retsch, Haan, Germany), using 2 ml tubes in a 96-well holder, each containing 2 stainless steel balls. Cysts of nematodes were manually picked from wet filter paper on which cysts and soil debris were collected after the described extraction. Two cysts were used for each RNA extraction. Approximately 10 µg of soft necrotic tissue was added directly to NucleoMag RNA kit MR1 buffer. The pathogen *Erwinia amylovora* was isolated from ca. 5 g of symptomatic tissue in a 25 mM PBS buffer (pH 7) by shaking at 1'000x rpm for 15 min at room temperature. DNA was then extracted from this solution (see below). For the fungal samples, ca. 0.1 g of symptomatic tissue was added to 50 µl extraction buffer, shaken at 15 min 1000 rpm 99˚C, followed by addition of 50 µl dilution buffer from the Extract-N-AMP Plant Tissue Kit (Sigma-Aldrich, St. Louis, MO, USA).

## Nucleic acid extraction, whole genome amplification and sequencing

Nucleic acid extraction was performed with various extraction kits (S1 Table) on a epMotion 5075t pipetting robot (Vaudaux-Eppendorf, Schönenbuch, Switzerland), using a custom-made workflow (available upon request). The workflow follows the main workflow proposed by the manufacturer with the following modifications: To accommodate microplate capacities, reagent volumes were reduced to 33–22% of those recommended, except for the elution step where the minimal recommended amount of 50 µl was used (details are summarized in S2 Table). Importantly, for RNA extractions the DNA digestion step was omitted to optimize total yield of nucleic acids. For RNA samples, cDNA production was achieved with the LunaScript™ RT SuperMix Kit (NEB, Ipswich, MA, USA) using 16 µl of the extracted nucleic acid and 4 µl of the kit mixture and following the manufacturer's instructions. Whole genome amplification (WGA) was performed with the GenomePlex Complete Whole Genome Amplification Kit WGA2 (Sigma-Aldrich, St. Louis, MO, USA), using 10 µl per sample of the cDNA corresponding to ca. 10–30 ng of cDNA and 17 cycles in the amplification step. Amplification products were cleaned using the MinElute PCR Purification Kit (QIAGEN, Hilden, Germany) with an elution volume of 12 µl. DNA end preparation was performed for 8 samples in parallel (ca. 15 ng DNA each) with a second automated workflow on the epMotion 5075t (available upon request) using the NEBNext Ultra II End Repair/dA-Tailing Module. Barcode ligation was again performed for the 8 samples in parallel with a third automated workflow on the epMotion 5075t (available upon request) using the Native Barcoding Expansion 1–12 Kit EXP-NBD104 (Oxford Nanopore Technologies, Oxford, UK). The individual barcode ligation products were quantified on a QuBit v.3 fluorometer (ThermoFisher, Waltham, MA, USA). The Sequencing library was prepared with the Ligation Sequencing Kit SQK-LSK109 (Oxford Nanopore Technologies, Oxford, UK). This step was also implemented as a fourth workflow on the epMotion 5075t (available upon request) starting with a single pool composed of equimolar contributions from the individual barcoded samples. The final libraries were then sequenced on a R.9.4.1 MinION flow cell with high accuracy base calling. After each run the flow cells were washed with the Wash Kit EXP-WSH003 (Oxford Nanopore Technologies, Oxford, UK), using the kit's storage buffer that enables reuse of the flow cells. The flow cells were stored at 4˚C until the next run.

## Data processing

To enable real time data processing, the MinION runs were base called with a Dell Precision 7920 Tower XCTO Base computer with two Intel Xeon Gold 6244 3.6GHz, 4.4GHz Turbo, 8C,

10.4GT/s 3UPI, 24.75MB Cache, HT (150W) DDR4-2933, 256GB (16x16GB) DDR4 2666MHz RDIMM ECC, a NVIDIA Quadro RTX6000, 24 GB, 4DP graphics card and 10TB hard disk space.

To perform base calling using GPU we used the software Guppy v. 4.5.4 with a parameter set established by Miles Benton (https://gist.github.com/sirselim/2ebe2807112fae93809 aa18f096dbb94). Passed fastq files of each individual barcode (deposited on SRA #PRJNA783774) were directly loaded into a separate folder in the software Geneious Prime v.2020.1.1 and processed with a custom made workflow performing the following tasks: 1) reads are mapped to a reference database (details see below) to produce contigs containing variable numbers of reads. Mapping was performed using the program minimap2 v.2.17 as implemented in the software Geneious Prime, with standard settings. 2) Among the 30 contigs with highest percentage of the length of the reference sequence covered by one or more reads, the one with the highest number of reads was further processed. Single base inserts occurring in less than 35% of reads were eliminated using the "Mask Alignment" function of Geneious Pro and a consensus sequence with these sites stripped based on the majority rule (most ambiguities) was produced and trimmed to the length of the reference sequence. 3) This consensus sequence is BLASTed (megablast) to a local copy of the NCBI nucleotide database (downloaded on 31.03.2021). The 10 top hits are then listed as the result in a new subfolder. Local BLAST was installed and performed within Geneious Prime using the nucleotide (nt) database downloaded from NCBI on May 10, 2020.

## Reference databases

The general insect reference database was established by downloading all open access fasta entries of Coleoptera, Diptera, Hemiptera, Homoptera, Hymenoptera, Lepidoptera and Thysanoptera for the marker COI (retaining only entries covering the 5' barcoding region) available on the Barcoding of Life Database (BOLD; http://www.barcodinglife.org/) on March 7th, 2020. An additional set of sequences specifically addressing coccoidea (Hemiptera) COI sequences that were underrepresented in the BOLD database was downloaded from GenBank (14.01.2021), again retaining only entries covering the 5' barcoding region) and merged with the main reference database. Furthermore, specific reference databases were established to test the feasibility of the workflow for species identification of nematodes, storage rot fungi, and bacteria. For identification of nematodes, the COI barcode sequences for Tylenchida were downloaded from GenBank (09.09.2020). For identification of storage rot fungi, the public ITS reference sequences available on BOLD were downloaded (23.06.2020). For identification of bacteria, the "Mini-barcode query sequence sets of bacterial/archaeal 16S" [20] were used as a reference database for identification of *Escherichia*, whereas the SILVA_138.1_LSURef_NR99_tax_silva database (20.10.2020) was used to for identification of *Erwinia*. For all databases, duplicate reads were removed using the "dedupe" V.38.37 (Brian Bushnell, https://github.com/BioInfoTools/BBMap/blob/master/sh/dedupe.sh) function implemented in Geneious Prime. Furthermore, reads sharing an identity higher than 97% were removed from the datasets. This enabled merging more reads into individual contigs and therefore improving to the quality of the resulting consensus sequences.

## Using additional genetic information

The mitogenomes of the species of the *Bactrocera dorsalis* species complex available on GenBank by October 2021 were downloaded and aligned in Geneious Prime: five of *Bactrocera dorsalis* (DQ917577, MG916968, KT343905, MN104220, DQ845759, KM244662); one each of *B. papayae* (DQ917578), *B. philippinensis* (DQ995281), and *B. invadens* (KX534207), and five

of *B. carambolae* (MG916967, NC009772, MN104219, MN104218, MN104217). A phylogenetic tree using FastTree v. 2.1.11 with default parameters as implemented in Geneious Prime was constructed to assess the correctness of species identification.

The full mitogenome coverage with samples for which more than 1.7 Mio reads were generated was assessed by mapping to reference mitogenomes downloaded from GenBank with minimap2 v.2.17 with default options for Oxford Nanopore reads for four species: *Bactrocera dorsalis* (MG916968); *Liriomyza sativae* (JQ862475); *Liriomyza trifolii* (JN570506) and *Drosophila suzukii* (KU588141).

The presence of reads of nuclear genes of interest, such as those known to contribute to pesticide resistance in diverse pest insects, among the sequenced genomic reads was assessed using the workflow with the following custom-made databases composed of reference sequences downloaded from GenBank in October 2021. For *Bactrocera dorsalis* these were acetylcholine esterase, fenitrothion resistant acetylcholine esterase, alpha esterase, carboxylesterase E4, carboxylesterase E6, glutathione-S-transferase, and internal transcribed spacer; for *Liriomyza sativae* the voltage-gated sodium channel gene; and for *Drosophila suzukii* Rho1, glutathione-S-transferase, and again voltage-gated sodium channel genes. Furthermore, five single copy nuclear genes (beta actin, arginine kinase, elongation factor alpha, glyceraldehyde-3-phosphate dehydrogenase, alpha tubulin [21]; were also searched the same way. In addition, as a proof-of-concept, mapping was also performed to search for endosymbiotic bacteria and for viruses in some RNA extracted samples.

## Results

In the process of developing the workflow we performed a total of 25 MinION (Mk1B) runs with single or multiple individually barcoded samples (range 1–8) on a total of 20 flow cells (R.9.4.1) (S1 Table). Some flow cells were used repeatedly for up to seven different library injections, in one case spanning a period of 77 days between the first and the last run.

### Run time and the use of pipetting robotics

The entire workflow takes ca. 12 hours including ca. three hours of hands-on time. Using pipetting robots for the nucleic acid extraction and sequencing library preparation leads to a considerable reduction of hands-on times by roughly 30%.

### Number of reads needed for identification

In general, we found a clear positive correlation between the number of reads and the quality of the consensus sequence assembled from them (S1 Fig). Our empirical data showed that 200'000 reads per sample generally resulted in a good performance of the workflow with highly accurate identifications in most cases (Table 1 and S1 Table). This keeps run-times per individual sample for a typical flow cell under one hour and safeguard the nanopores of the flow cells for follow-up runs.

### Sample identification

Overall, pairwise identity between each individual sample consensus sequence (N = 187'153.6 ± 31'374.0 reads, average ± standard deviation; median = 199'547.5) and the best match of the GenBank BLAST averaged 99.6 ± 0.6% (median = 99.8%) (Table 1). For most of the samples, assumed species based on morphology could be confirmed while for some of the samples (aphids, Tenthredinidae, whiteflies) could be assigned to a deeper

**Table 1. Summary statistics for the workflow for nanopore sequencing based species identification within each of five orders of insects.**

| | | N reads | Sequence Length | % Pairwise ID | Bit-Score | E-Value | % Query Coverage | N reads assembled to best hit ref (Max#seqs) | N reads mapped to refbase | N contigs produced of mapped reads | N reads for 50x coverage of best hit | Ratio of reads mapped to best hit vs reads mapped to refbase |
|---|---|---|---|---|---|---|---|---|---|---|---|---|
| Diptera N[a] = 31 | AVG | 187859 | 656 | 99.7 | 1195.2 | 0.0 | 99.3 | 65.2 | 508 | 74 | 309119.9 | 0.80 |
| | STD | 27970 | 5 | 0.5 | 49.4 | 0.0 | 2.7 | 59.5 | 976 | 80 | 383209.9 | 0.37 |
| | Median | 200014 | 658 | 99.8 | 1208.8 | 0.0 | 100.0 | 45.0 | 80 | 48 | 194423.0 | 0.98 |
| | Min | 104065 | 632 | 97.4 | 946.6 | 0.0 | 85.2 | 6 | 14 | 9 | 39172 | 0.00 |
| | Max | 217798 | 658 | 100.0 | 1216.2 | 0.0 | 100.0 | 278 | 2887 | 342 | 1671675 | 1.09 |
| Hemiptera N = 18 | AVG | 178246 | 576 | 99.8 | 1056.0 | 0.0 | 99.1 | 109.9 | 99 | 16 | 172446.3 | 1.09 |
| | STD | 38119 | 50 | 0.3 | 83.8 | 0.0 | 3.2 | 70.0 | 65 | 10 | 212835.0 | 0.13 |
| | Median | 198219 | 543 | 100.0 | 1003.9 | 0.0 | 100.0 | 111.5 | 92 | 16 | 74289.5 | 1.08 |
| | Min | 93779 | 543 | 98.9 | 996.5 | 0.0 | 86.1 | 11 | 14 | 1 | 40989 | 0.79 |
| | Max | 225139 | 658 | 100.0 | 1203.3 | 0.0 | 100.0 | 248 | 243 | 38 | 902064 | 1.31 |
| Hymenoptera N = 8 | AVG | 183461 | 624 | 99.6 | 1132.2 | 0.0 | 94.7 | 66.9 | 928 | 64 | 420494.5 | 0.52 |
| | STD | 24019 | 67 | 0.3 | 134.4 | 0.0 | 10.2 | 81.0 | 1435 | 46 | 329237.8 | 0.41 |
| | Median | 192003 | 657 | 99.5 | 1190.4 | 0.0 | 99.8 | 28.5 | 101 | 43 | 367954.0 | 0.51 |
| | Min | 124947 | 453 | 98.9 | 789.6 | 0.0 | 68.7 | 11 | 47 | 14 | 32192 | 0.00 |
| | Max | 201053 | 658 | 100.0 | 1216.2 | 0.0 | 100.0 | 265 | 3692 | 142 | 910091 | 1.06 |
| Lepidoptera N = 8 | AVG | 206695 | 602 | 98.9 | 1066.2 | 0.0 | 91.4 | 9.6 | 14 | 44 | 1281512.1 | 0.74 |
| | STD | 25492 | 114 | 0.8 | 204.7 | 0.0 | 17.2 | 3.4 | 6 | 23 | 665535.9 | 0.15 |
| | Median | 218195 | 644 | 99.2 | 1144.2 | 0.0 | 97.9 | 10.5 | 13 | 42 | 948515.5 | 0.73 |
| | Min | 140844 | 302 | 97.4 | 538.5 | 0.0 | 46.1 | 4 | 6 | 20 | 783511 | 0.54 |
| | Max | 219424 | 658 | 99.9 | 1212.5 | 0.0 | 100.0 | 14 | 26 | 86 | 2742800 | 1.00 |
| Thysanoptera N = 1 | AVG | 198844 | 644 | 99.5 | 1170.1 | 0.0 | 98.9 | 22.0 | 19 | 4 | 451918.0 | 1.16 |
| | STD | 0 | 0 | 0.0 | 0.0 | 0.0 | 0.0 | 0.0 | 0 | 0 | 0.0 | 0.00 |
| | Median | 198844 | 644 | 99.5 | 1170.1 | 0.0 | 98.9 | 22.0 | 19 | 4 | 451918.0 | 1.16 |
| | Min | 198844 | 644 | 99.5 | 1170.1 | 0.0 | 98.9 | 22 | 19 | 4 | 451918 | 1.16 |
| | Max | 198844 | 644 | 99.5 | 1170.1 | 0.0 | 98.9 | 22 | 19 | 4 | 451918 | 1.16 |
| ALL N = 66 | AVG | 187154 | 624 | 99.6 | 1133.6 | 0.0 | 97.7 | 70.2 | 380 | 53 | 405374.5 | 0.84 |
| | STD | 31374 | 63 | 0.6 | 119.7 | 0.0 | 7.9 | 68.7 | 884 | 63 | 510305.2 | 0.36 |
| | Median | 199548 | 658 | 99.8 | 1190.4 | 0.0 | 100.0 | 44.5 | 69 | 31 | 196287.5 | 1.00 |
| | Min | 93779 | 302 | 97.4 | 538.5 | 0.0 | 46.1 | 4 | 6 | 1 | 32192 | 0.00 |
| | Max | 225139 | 658 | 100.0 | 1216.2 | 0.0 | 100.0 | 278 | 3692 | 342 | 2742800 | 1.31 |

[a]N = Number of analyzed specimens within order.

taxonomic level with the novel approach while some of the samples (Bactrocera, Delia) resulted in a different species than initially assumed based solely on morphology.

The workflow also proved to be suitable for species identification in other animal classes and kingdoms. We successfully identified species of bacteria based on 16S rDNA, of nematodes based on COI and of storage rot fungi based on ITS sequences (Table 2) using the appropriate reference database.

## Adding alternative genetic identification markers

The whole genome based strategy chosen for this workflow enables to use any genetic marker for which reference data are available. This is especially true for all mitochondrial genes that occur in high copy numbers, as, in addition to COI, other mitochondrial genes may be

**Table 2. Molecular species diagnosis of bacteria, storage rot fungi and a nematode using the workflow with specific target reference databases.**

| Species | Gene | Mapped to Reference Sequence | N reads analyzed | Sequence Length | % Pairwise ID | Bit-Score | E-Value | % Query Coverage | N reads mapped to refbase | Max reads in contig | Best Hit | Remarks | SRA file name |
|---|---|---|---|---|---|---|---|---|---|---|---|---|---|
| Bacteria Escherichia coli | 16S rDNA | RefDB "Tanabe13_bacteria_16S_species"; 338405 sequences | 419526 | 804 | 99.9 | 1478.4 | 0 | 99.6 | 5099 | 366 | CP044293.1 | Escherichia coli strain P276M | 210218_Bacteria_Ecoli |
| Bacteria: Erwinia amylovora | rDNA | SILVA_138.1_LSURef_NR99_tax_silva_dedupe97 | 198709 | 2936 | 99.9 | 5398.9 | 0 | 100.0 | 6720 | 242 | FN666575.1 | Erwinia amylovora ATCC 49946 | 201008_Erwinia_TP |
| Fungi: Neofabraea | I T S | BOLD_Fungi_200623 | 2E+06 | 877 | 99.9 | 1615.1 | 0 | 100.0 | 51646 | 872 | FJ654654 | Neofabraea alba | 200616_RNA_Fungi_buan/ barcode04 |
| Fungi: Cladosporium | I T S | BOLD_Fungi_200623 | 310751 | 2851 | 99.7 | 5193.9 | 0 | 99.1 | 13895 | 422 | MH047193 | Cladosporium herbarum | 200616_RNA_Fungi_buan/ barcode03 |
| Fungi: Colletotrichum | I T S | BOLD_Fungi_200623 | 424457 | 1626 | 98.9 | 2885.6 | 0 | 99.6 | 20364 | 393 | MK541032 | Colletotrichum salicis isolate RB157 | 200616_RNA_Fungi_buan/ barcode02 |
| Fungi: Colletotrichum | I T S | BOLD_Fungi_200623 | 424457 | 1626 | 98.9 | 2885.6 | 0 | 99.6 | 20364 | 393 | MK541032 | Colletotrichum salicis isolate RB157 | 200616_RNA_Fungi_buan/ barcode02 |
| Nematode: Globodera rostochiensis | I T S | GenBank_Tylenchida_COI_200909 | 181271 | 443 | 100 | 819.2 | 0 | 100.0 | 126 | | MT240262 | Globodera rostochiensis NRM67 | 15_200909_Globros_BC04 |

mapped to improve species discrimination. In fact, if more than ca. 1.5 Mio reads are available, it is possible to assemble the entire mitogenome (S3 Table).

### Single copy nuclear genes and genes of interest

The sequencing strategy developed in this study also allows searching for single copy nuclear genes or other genes of interest not necessarily linked to species molecular identification. For example, genes that are involved in pesticide resistance in a number of insects, such as sodium channel genes, various esterases and glutathione-S-transferases, will be available for analysis, provided enough reads are collected. All five single copy nuclear genes that were searched by mapping to the corresponding reference sequences in each of the three Diptera species for which more than 0.5 Mio reads were collected, i.e., *Bactrocera dorsalis*, *Drosophila suzukii* and *Liriomyza sativae* could be confirmed in all species (S4 Table). In addition, all genes of interest, e.g., genes putatively involved in pesticide resistance or internal transcribed spacer sequences, and their sequence could be extracted in *Bactrocera dorsalis*, *Liriomyza sativae* and *Drosophila suzukii* (S5 Table).

### Exploiting the full potential of whole genome information

In addition to enable using more than a single genetic marker for species identification, our strategy also enables to search in the genomic data for bacterial endosymbionts and DNA viruses, or, if RNA was extracted from the sample, arthropod borne pathogenic RNA viruses. However, as with single copy nuclear genes, this requires collecting higher numbers of reads per sample. For example, based on mapping 1'707'634 reads of *Drosophila suzukii* sample 08_200611_Drossuz_BC12 to a custom made diptera virus reference database containing 276 viruses (GB_Diptera_Virus_200505), an 843 bp fragment composed of 64 reads with a 98.4% pairwise identity to Teise virus isolate UK1 (MF893269.1) was found. This virus was reported before to occur at relatively high titers *in D. suzukii* [22]. Similarly, mapping 576749 reads from a whitefly larva (sample 15_200909_Triavap_BC05_500k) to a reference database of 1200 sequences of the widespread insect endosymbiotic bacteria genus Buchnera (GenBank download on September 25 2020) produced a 1132 bp long contig composed of 132 reads, that upon BLAST, resulted in a 99.21% identity to *Candidatus* Portiera aleyrodidarum (Z11928.1), the primary endosymbionth of *Trialeurodes vaporariorum*.

## Discussion

Nanopore technology is building up momentum as a new diagnostics tool [23–26] however; it is far from having reached its full potential. One of the main reasons may be the lack of robust and easy to use workflows. Here we present a workflow for genome based nanopore diagnostics of insects that, using the classical DNA barcoding fragments as references, enables reliable identifications to the species level of many agronomically relevant pests or pathogens. In addition, our workflow also eliminates developing specific primers for the amplification of the barcoding fragment and is therefore robust towards allelic dropouts. Because the mitochondrial genome occurs in many copies per cell [27], robust identification results for insects can be obtained with a relatively low number of only ca. 200'000 genomic reads per sample. In terms of sequencing time, this typically corresponds to one hours of sequencing on a MinION flow cell. By using a mean of 300'000 reads, the coverage of the best hit reached 50x, which relates to an estimated consensus accuracy of Q50, thus excluding errors almost entirely. The need for this amount of reads will decrease in the near future with ONTs current chemistry promising a Q50 consensus quality with only 20x coverage, reducing the number of reads required to well below the 200'000 used here, possibly to as low as 120'000 (https://nanoporetech.com/

accuracy). The entire workflow described here lasts around 12 hours including approximately three hours hands-on time, depending on whether pipetting robots are used.

Since reads are collected across the entire genome, other mitochondrial gene fragments may also be used. In fact, if more than ca. 1.5 Mio reads are available, it is possible to assemble the entire mitogenome (S3 Table). For example, the Tephritid species *Bactrocera dorsalis* encompasses a group of very closely related Tephritids that cannot be differentiated based on the classical DNA barcoding fragment alone [15]. However, applying all Tephritid mitogenomes, it appears to be possible differentiating *B. carambolae* from the rest of the species group by as much as 12 species-specific diagnostic single nucleotide polymorphisms. This differentiation is even more pronounced if using the ribosomal internal transcribed spacer region. Since the ribosomal genes are also present in multi-copies [28], the internal transcribed spacer regions 1 and 2 (ITS1, ITS2) can easily be recovered and provide additional phylogenetic information (see S5 Table, 1'018 of 507'790 reads mapped to the rDNA reference of *B. dorsalis*).

Using ITS as a supplement to COI and possibly other mitogenomic gene regions is a promising strategy for accurate species diagnostics in future improvements of this workflow.

In addition, the workflow presented here also allows searching for nuclear genes of interest (S5 Table) as well as for the eventual occurrence and identification of arboviruses and/or prokaryotic endosymbionts as mentioned in the results section above. This allows answering questions with relevance beyond simply identifying the species, such as the identification of resistance genes informing efficient selection of plant protection antibiotics or fungicides as in apple fire blight [29, 30] or apple scab prevention [31]. Similarly, the presence of previously described insecticide resistance genes can inform the efficient application of insecticides [32, 33]. Additionally, proposed arbovirus surveillance schemes using deep sequencing [34] could profit from the "sequence anywhere" capability of Oxford Nanopores sequencing platform.

The workflow thus demonstrates its suitability for COI-based DNA barcoding identification of insects and nematodes, for ITS based identification of fungi and 16S rDNA based identification of bacteria with high accuracy while at the same time offering the possibility to extend beyond using single genes for diagnostics, thereby enabling genome based specimen identification and genetic characterization. The generic mode of nucleic acid exploration applied here could be used in all organisms for which the main genetic content of the sample originates from the target organism, i.e. in fungal and bacterial cultures and possibly enriched extract fractions of pathogenic viruses. The presented work has explicitly been developed for the identification of single unknown individuals. While the Nanopore sequencing approach can be applied to research questions where mixtures of species are present a completely different bioinformatics approach would be chosen similar to a metagenomic sequence classification as in Kraken [35].

Overall, the developed workflow allows (i) to successfully processing insect species for which it is difficult to obtain amplicons due to missing primer sequence information, (ii) to assess the presence of genes of interest such as insecticide resistance genes, and (iii) to monitor the presence of arthropod-infecting viruses and endosymbiotic bacteria. In future work, the generated dataset may be also exploited for insect dietary analyses [36] enabling the collection of information about potential host-plants of the studied specimen. Finally, provided the necessary reference databases will be developed and appropriate read numbers can be sampled, the workflow may furthermore enable epidemiological analyses to reveal the origin and possible invasion pathway of introduced pests and pathogens [37, 38]. Such information is key to implement control measures such as quarantine or eradication preventing their further spread [38].

## Supporting information

**S1 File. Step-by-step protocol using the MonarchGenomic DNA Purification Kit (New England Biolabs NEB, Ipswich, MA, USA) downloaded from protocols.io.** dx.doi.org/10.17504/protocols.io.bx7nprme.
(PDF)

**S1 Table. Identification of 66 individual insects covering 29 species of 13 families and 5 orders.**
(XLSX)

**S2 Table. Adaptations to the original Macherey-Nagel protocol.**
(XLSX)

**S3 Table. Summary of mitogenome assemblies established from mapping reads to the respective reference genomes for four species of Diptera.** It should be noted that the mitogenome sequences do not include the hypervariable D-Loop region.
(XLSX)

**S4 Table. Mapping single copy nuclear genes in three Diptera species for which more than 0.5 Mio reads were sequenced.**
(XLSX)

**S5 Table. Mapping genes of interest (e.g., genes often involved in pesticide resistance and internal transcribed spacer sequences) in three Diptera species for which more than 0.5 Mio reads were sequenced.**
(XLSX)

**S1 Fig. Correlation between the number of collected reads used for the mapping analysis and the number of hits mapping to the reference database.** Included are samples for which at least 500'000 reads were sampled and more than four mapping reads were found: *Bactrocera dorsalis*, *Ceratitis capitata*, *Drosophila suzukii*, *Helicoverpa armigera*, *Liriomyza sativae*, *Liriomyza trifolii*, *Spodoptera exigua*, *Trialeurodes vaporariorum*, *Zeugodacus cucurbitae*.
(TIF)

## Acknowledgments

We would like to thank Maja Hilber, Agroscope, for providing the sample of the bacterial disease *Erwinia amylovora* and Paul Dahlin, Agroscope, for providing the nematode pathogen sample of *Globodera rostochiensis*.

## Author Contributions

**Conceptualization:** Jürg E. Frey, Simon Blaser, Morgan Gueuning, Andreas Bühlmann.

**Data curation:** Jürg E. Frey.

**Formal analysis:** Jürg E. Frey.

**Methodology:** Jürg E. Frey, Beatrice Frey, Daniel Frei, Simon Blaser, Morgan Gueuning, Andreas Bühlmann.

**Visualization:** Morgan Gueuning.

**Writing – original draft:** Jürg E. Frey.

**Writing – review & editing:** Jürg E. Frey, Simon Blaser, Morgan Gueuning, Andreas
Bühlmann.

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
