## [Decision Letter · Decision Letter 0]

24 Mar 2022

PONE-D-22-03204Next generation biosecurity: Towards genome based identification to prevent spread of agronomic pests and pathogens using nanopore sequencingPLOS ONE

Dear Dr. Buehlmann,

Thank you for submitting your manuscript to PLOS ONE. After careful consideration, we feel that it has merit but does not fully meet PLOS ONE’s publication criteria as it currently stands. Therefore, we invite you to submit a revised version of the manuscript that addresses the points raised during the review process.

Please submit your revised manuscript by  May 8 2022. If you will need more time than this to complete your revisions, please reply to this message or contact the journal office at plosone@plos.org. Please include the following items when submitting your revised manuscript:A rebuttal letter that responds to each point raised by the academic editor and reviewer(s). You should upload this letter as a separate file labeled 'Response to Reviewers'.A marked-up copy of your manuscript that highlights changes made to the original version. You should upload this as a separate file labeled 'Revised Manuscript with Track Changes'.An unmarked version of your revised paper without tracked changes. You should upload this as a separate file labeled 'Manuscript'.

We look forward to receiving your revised manuscript.

Kind regards,

Patrizia Falabella

Academic Editor

PLOS ONE

Journal Requirements:

2. We noted in your submission details that a portion of your manuscript may have been presented or published elsewhere. [This submission is part of the protocols.io/PLoS one collaboration. The corresponding protocols.io document is available here:

dx.doi.org/10.17504/protocols.io.bx7nprme

The materials & methods section is very similar to the one on protocols.io but the intro/results/discussion are only presented here]  Please clarify whether this publication was peer-reviewed and formally published. If this work was previously peer-reviewed and published, in the cover letter please provide the reason that this work does not constitute dual publication and should be included in the current manuscript.

Reviewers' comments:

Reviewer's Responses to Questions

**Comments to the Author**

1. Does the manuscript report a protocol which is of utility to the research community and adds value to the published literature?

Reviewer #1: Yes

Reviewer #2: Yes

2. Has the protocol been described in sufficient detail?

Descriptions of methods and reagents contained in the step-by-step protocol should be reported in sufficient detail for another researcher to reproduce all experiments and analyses. The protocol should describe the appropriate controls, sample sizes and replication needed to ensure that the data are robust and reproducible.

Reviewer #1: Yes

Reviewer #2: Yes

3. Does the protocol describe a validated method?

Reviewer #1: Yes

Reviewer #2: Yes

4. If the manuscript contains new data, have the authors made this data fully available?

Reviewer #1: Yes

Reviewer #2: Yes

**5. Is the article presented in an intelligible fashion and written in standard English?**

Reviewer #1: Yes

Reviewer #2: Yes

6. Review Comments to the Author

Reviewer #1: ID PONE-D-22-03204: “Next generation biosecurity: Towards genome based identification to prevent spread of agronomic pests and pathogens using nanopore sequencing” by Frey et al.

The manuscript reports a protocol that can be useful to the research community interested in the identification of agronomic pests. The traditional methods are based on classical barcoding (or metabarcoding) that targets one or a few genomic regions of one or a small group of species. The barcoding approach requires some prior knowledge of the target organisms, which sometimes is lacking. In this manuscript, the authors take the metagenomic approach by sequencing all the DNA of the sample, so any genomic region of any organism can be sequenced and, if there is enough genomic information in the repositories, identified. In consequence, the proposal is more flexible than the classical one based on barcoding. I reckon that in the future (with richer genomic repositories) methods like the one proposed here based on metagenomics will overcome (meta)barcoding methods.

The protocol is described in sufficient detail in the main text and the supplementary material. The method is validated using more than 60 artificial single-species libraries of insects and a few more of bacteria, fungi, and nematodes.

Despite being very positive about the manuscript, I don’t like how results are presented. I have a couple of suggestions that may be useful to the authors in the preparation of a new version of the manuscript.

Perhaps Figure 1 is unnecessary. It is just to be expected to find a highly significant linear relationship between the number of mapping reads and the number of collected reads. This information can be included in the text.

On the contrary, I’ve found the description of sample identification too shallow (l. 231-237). The authors only provide summary statistics (Table 1 and 2) that show that the process of species identification is good, in general. However, this is not always the case, as there are 9 libraries in which the “assumed species” is different to the “species ID” (Supplementary Table 2: please clarify the meaning of these two columns). This criticism does not invalidate the method but is proof that the method is not perfect (as it happens with all DNA-based technologies for species identification). I encourage the authors to acknowledge the limitations of their method in the main text (in the results) and explain, if possible, the reason for the wrong or partial identification of some species (in the discussion).

A final comment about the application of the method to real samples. The manuscript uses artificial single species libraries and the bioinformatic pipeline is intended to provide one species per sample. But what would happen if the original sample contained more than one insect species? Ideally, the method should provide all of them, but would it? In addition, if the sample contained two phylogenetically close species (e.g. two species of Bactrocera) would their reads map in different contigs or would they be mixed? I understand that these questions can be difficult to handle at this stage, but the authors might consider adding their reflections to the discussion. Depending on their willingness, they could simulate two-species libraries by merging the fastq files of two single-species libraries, run the entire pipeline using the new fastq file, and see what happens.

Reviewer #2: This manuscript is introducing a WGA based protocol utilizing ON MinION sequencing to improve DNA-based species identification. the methods is described very well, sufficient details on all steps are provided. My major point of criticism is that the discussion section lacks several aspects mentioned in the results e.g., on the presence of genes of interest such as insecticide resistance genes, and the presence of arthropod-infecting

viruses and endosymbiotic bacteria. for a protocol description such as this it would be also very interesting to discuss the reference libraries used – where they sufficient? If not, how to improve that, where are the gaps, etc. ; or what about the pipelines you used? Where they useful, what could be alternatives also given that Geneious is not cheap?

Minor things:

L29 - 658 basepairs if you refer to COI

L58 - is needed not are needed

L64 - shown not proofed

L125 - add a 'the' between extractions and DNA

L128 - add the missing )

L214 – How did you compensate for retained sequences of prior runs? MinION flow cells are known to retain up to 15% of former runs even if their wash kits are used. Did you see a difference (or overlap) between runs of reused flowcells? This should also be briefly discussed.

L239 – table caption talks about Nematode data, but none is shown.

L309 – remove “should essentially enable to” and use ‘could’

L314 – Interest not interests

7. PLOS authors have the option to publish the peer review history of their article (what does this mean?). If published, this will include your full peer review and any attached files.

Reviewer #1: No

Reviewer #2: No

---

## [Author Response · Author response to Decision Letter 0]

14 Apr 2022

Responses to all issues raised are in the file "Responses to Reviewers.docx"

---

## [Decision Letter · Decision Letter 1]

20 Jun 2022

Next generation biosecurity: Towards genome based identification to prevent spread of agronomic pests and pathogens using nanopore sequencing

PONE-D-22-03204R1

Dear Dr. Buehlmann,

We’re pleased to inform you that your manuscript has been judged scientifically suitable for publication and will be formally accepted for publication once it meets all outstanding technical requirements.

Kind regards,

Patrizia Falabella

Academic Editor

PLOS ONE

Additional Editor Comments (optional):

Reviewers' comments:

Reviewer's Responses to Questions

**Comments to the Author**

1. Does the manuscript report a protocol which is of utility to the research community and adds value to the published literature?

Reviewer #1: Yes

2. Has the protocol been described in sufficient detail?

Descriptions of methods and reagents contained in the step-by-step protocol should be reported in sufficient detail for another researcher to reproduce all experiments and analyses. The protocol should describe the appropriate controls, sample sizes and replication needed to ensure that the data are robust and reproducible.

Reviewer #1: Yes

3. Does the protocol describe a validated method?

Reviewer #1: Yes

4. If the manuscript contains new data, have the authors made this data fully available?

Reviewer #1: Yes

**5. Is the article presented in an intelligible fashion and written in standard English?**

Reviewer #1: Yes

6. Review Comments to the Author

Reviewer #1: This is the second time that I review this manuscript. As in this first occasion, I am positive about its quality. The protocol is described in sufficient detail in the main text and the supplementary material, and the method is validated using artificial single-species libraries. The suggestions that I made in my previous review are satisfactorily considered in the new version, so I almost have nothing else to add.

Just one word of caution to the suggestion that Kraken is a good alternative to their bioinfomatic approach to identify species in DNA libraries. Kraken, like many other metagenomic classifiers, is rather prone to produce false positives (see, for example, Ye SH, Siddle KJ, Park DJ, Sabeti PC. Benchmarking metagenomics tools for taxonomic classification. Cell. 2019;178(4):779-794).

7. PLOS authors have the option to publish the peer review history of their article (what does this mean?). If published, this will include your full peer review and any attached files.

Reviewer #1: No

---

## [Editor Report · Acceptance letter]

13 Jul 2022

PONE-D-22-03204R1 

Next generation biosecurity: Towards genome based identification to prevent spread of agronomic pests and pathogens using nanopore sequencing 

Dear Dr. Bühlmann:

I'm pleased to inform you that your manuscript has been deemed suitable for publication in PLOS ONE. Congratulations! Your manuscript is now with our production department. 

Kind regards, 

on behalf of

Prof. Patrizia Falabella 

Academic Editor

PLOS ONE